# Structure and analysis of nanobody binding to the human ASIC1a ion channel

Yangyu Wu[1], Zhuyuan Chen[1], Fred J Sigworth[2], Cecilia M Canessa[1,2]*

[1]Basic Sciences Department, Tsinghua University School of Medicine, Beijing, China; [2]Cellular and Molecular Physiology, Yale University School of Medicine, New Haven, United States

**Abstract** ASIC1a is a proton-gated sodium channel involved in modulation of pain, fear, addiction, and ischemia-induced neuronal injury. We report isolation and characterization of alpaca-derived nanobodies (Nbs) that specifically target human ASIC1a. Cryo-electron microscopy of the human ASIC1a channel at pH 7.4 in complex with one of these, Nb.C1, yielded a structure at 2.9 Å resolution. It is revealed that Nb.C1 binds to a site overlapping with that of the Texas coral snake toxin (MitTx1) and the black mamba venom Mambalgin-1; however, the Nb.C1-binding site does not overlap with that of the inhibitory tarantula toxin psalmotoxin-1 (PcTx1). Fusion of Nb.C1 with PcTx1 in a single polypeptide markedly enhances the potency of PcTx1, whereas competition of Nb.C1 and MitTx1 for binding reduces channel activation by the toxin. Thus, Nb.C1 is a molecular tool for biochemical and structural studies of hASIC1a; a potential antidote to the pain-inducing component of coral snake bite; and a candidate to potentiate PcTx1-mediated inhibition of hASIC1a in vivo for therapeutic applications.

**\*For correspondence:**
cecilia.canessa@yale.edu

**Competing interests:** The authors declare that no competing interests exist.

## Introduction

ASICs are proton-activated sodium channels present in most neurons of the central and peripheral nervous systems (*Krishtal and Pidoplichko, 1981*; *Waldmann et al., 1997*). There are four ASIC genes (*ASIC1-4*) and six isoforms in the human genome (*Kellenberger and Schild, 2002*). The most abundant and broadly expressed subunit is ASIC1a; its deletion in the mouse genome eliminates most of the proton-induced currents mediated by ASICs (*Wemmie et al., 2002*). Association of three pore-forming subunits, either as homotrimers or heterotrimers, forms functional channels. All ASIC isoforms share a common structure consisting of two transmembrane domains (TMDs) (TM1 and TM2), cytosolic amino- and carboxy-termini with the amino-terminus forming a reentrant loop into the lower pore (*Yoder and Gouaux, 2020*), and a large extracellular domain (ECD) that adopts a fist-like conformation with distinct subdomains. These structural features were unveiled by the first crystal structure (*Jasti et al., 2007*), which was obtained from chicken ASIC (cASIC1) in the desensitized conformation at low pH. Subsequently, structures of cASIC1 in the open (*Baconguis et al., 2014*), closed (*Yoder et al., 2018*), desensitized at high and low pH (*Gonzales et al., 2009*; *Yoder and Gouaux, 2020*), and in complex with various toxins have been resolved (*Dawson et al., 2012*; *Baconguis and Gouaux, 2012*; *Baconguis et al., 2014*). Recently, structures of a human ASIC subunit, hASIC1a, have been obtained alone and in complex with Mambalgin-1 toxin (*Sun et al., 2020*).

ASICs are the target of many polypeptide toxins that induce significant functional changes (*Cristofori-Armstrong and Rash, 2017*). Of the three most studied, two are antagonists and one is an activator of ASIC channels. PcTx1 is a 40-residue peptide from the venom of the tarantula *Psalmopoeus cambridgei* (*Escoubas et al., 2000*) that inhibits hASIC1a (IC$_{50}$ ~3 nM when conditioned at pH 7.2). It has been investigated as an analgesic and neuroprotective agent from ischemic injury of the brain (*Mazzuca et al., 2007*; *Xiong et al., 2004*). Mambalgin-1 is a 78-residue three-finger toxin

from the venom of the black mamba snake *Dendroaspis polypepsis* that rapidly and reversibly inhibits ASIC1a in neurons (*Diochot et al., 2012*). Because Mambalgin-1 exhibits strong analgesic effects, it has raised interest as a potential treatment for chronic pain and as an alternative to opioids (*Diochot et al., 2016*). Third, the toxin MitTx, found in the venom of the Texas coral snake *Micrurus tener tener*, functions as a potent, persistent, and selective agonist for ASICs (*Bohlen et al., 2011*). It is a heterodimer of an α-subunit (60 a.a.) and β-subunit (119 a.a.).

These toxins bind primarily to the thumb domain of the ECD of ASICs. Structures of each of these toxins in complex with cASIC1 have been solved, and in the recent study of hASIC1a by *Sun et al., 2020* a cryo-electron microscopy (cryo-EM) structure of hASIC1a bound to Mambalgin-1 at pH 7.4 was solved at 3.9 Å resolution.

To address present challenges in analysis of hASIC1a, we have developed nanobodies (Nbs) with high specificity and affinity to human ASIC1a. These versatile molecules are derived from the variable domain ($V_{HH}$) of single-domain antibodies produced in camelids (*Hamers-Casterman et al., 1993*) and in cartilaginous fishes (*Stanfield et al., 2004*). We present a set of Nbs for uses in novel applications that extend to the possibility of examining structure and function of hASIC1a in vivo and in vitro and as potential therapies of pathologic conditions involving hASIC1a.

## Results

### Generation of Nbs specific to human ASIC1a

An alpaca was immunized through the injection of 293 T cells expressing full-length hASIC1a on a 9-week immunization schedule (*Figure 1—figure supplement 1A*). Sera collected post-immunization contained conventional IgG1 and single-domain IgG2/3 antibodies that bind to hASIC1a, as analyzed by ELISA (*Figure 1A–B*). A phage display library was constructed from mRNA isolated from peripheral lymphocytes (titer of approximately $10^9$ independent clones) (*Pardon et al., 2014*; *Figure 1—figure supplement 1B–E*). To select Nbs that recognize hASIC1a in native conformation and to minimize isolation of nonspecific binders, three successive panning protocols were used: the first panning was conducted on *Xenopus* oocytes injected with hASIC1a cRNA while the second and third used recombinant hASIC1a protein bound to agarose and magnetic beads, respectively (*Figure 1C*). Phagemids recovered from the third panning were used for expression and isolation of Nbs: 600 were tested by ELISA. We considered a clone to be positive if the ELISA intensity was above a threshold that eliminated about two-thirds of the clones. Representative ELISA results are shown in *Source data 1*. Approximately 200 clones were selected for sequencing of DNA. The sequencing result indicated that many of ELISA positive clones were identical or had one or two amino acid differences, which is consistent with efficient enrichment for high affinity clones obtained by our screening strategy. DNA sequences of final candidates separated into three main groups, as shown in a phylogeny tree (*Figure 1E*) and in the protein alignment of *Figure 1—figure supplement 2*. All these clones were further examined by immunofluorescence of cells transfected with hASIC1a. From all the Nbs tested, the group consisting of C1-4-5, D10, and H10 produced strong signals and low background. A different group (1A-B1, 2B-D4, and 2B-E60) required permeabilization of cells for labeling, suggesting that the recognized epitopes are intracellular; this was confirmed in immunocytochemistry of permeabilized cells.

### Nb.C1 stabilizes and prevents aggregation of hASIC1a

With the exception of the one recent cryo-EM study of hASIC1a (*Sun et al., 2020*), all structural information about ASIC channels has been obtained from the chicken isoform. Although it shares approximately 89% sequence identity with hASIC1a, the chicken isoform differs in functional properties and response to toxins and other compounds (*Saez et al., 2011*; *Alijevic and Kellenberger, 2012*; *Smith and Gonzales, 2014*). Thus, when considering the development of therapeutics, the structure of the human channel is preferred. In the past, challenges working with hASIC1a arose from the tendency of hASIC1a to give low yields and to aggregate, making it difficult to obtain high-quality protein preparations suitable for structural analysis.

One of the goals of this work was to explore whether an Nb with high affinity to hASIC1a would overcome these problems. Among the best binders initially screened, Nb.C1was selected on the basis of high-affinity, low background, and absence of modification of channel function.

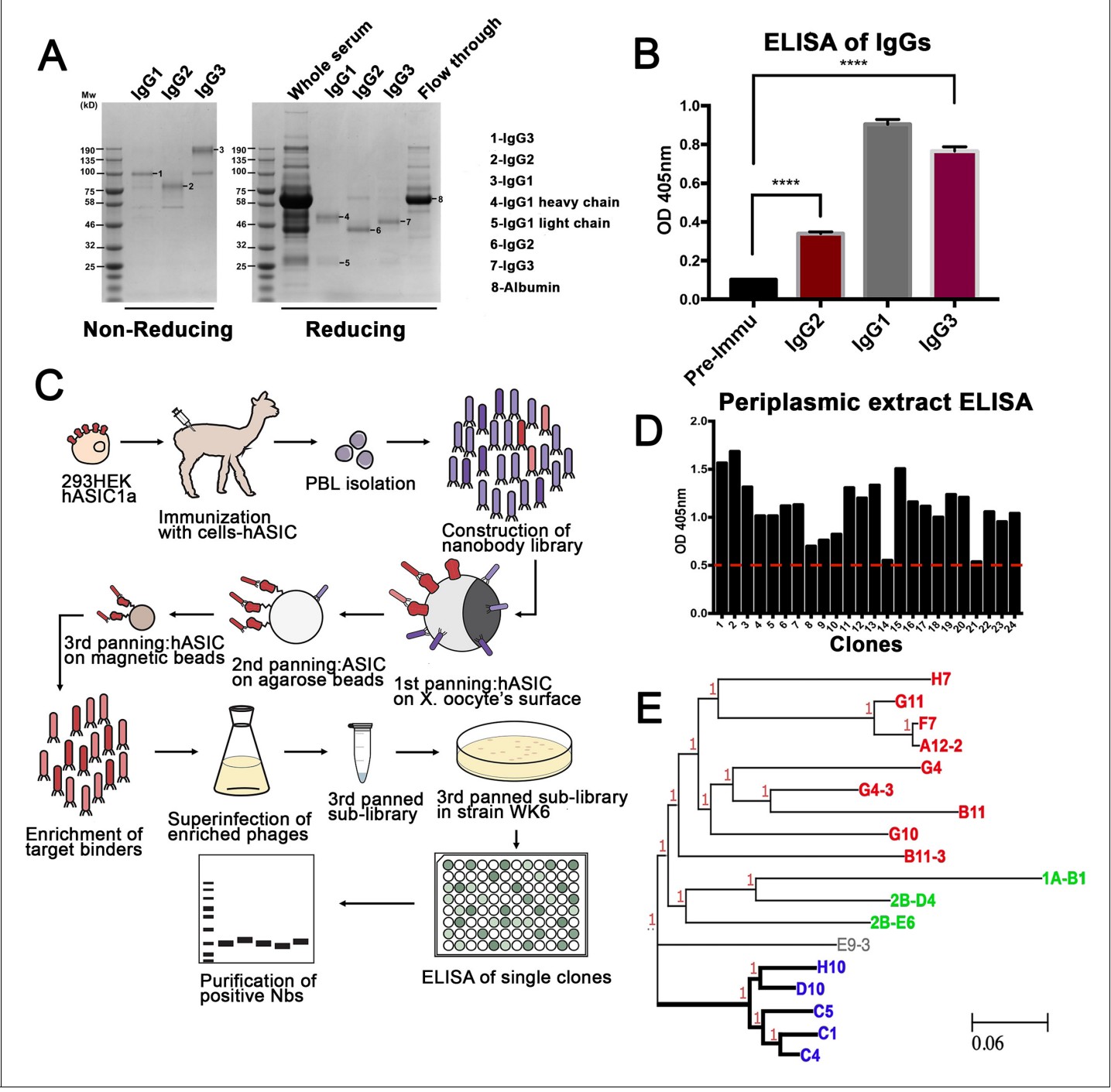

**Figure 1.** Generation of nanobodies (Nbs) specific to hASIC1a. (**A**) Fractionation of IgG (immunoglobulin G) classes from serum after completion of immunization schedule of alpaca. (**B**) Each fraction was tested for antibodies against hASIC1a by ELISA. All three immunoglobulin fractions, including single-domain antibodies Ig2 and IgG3 shown in red columns, are significantly higher than the pre-immune serum, t-test p-value < 0.001. (**C**) Overall method for generation of a phage display library, panning strategy for selection of highly reactive phages, and final purification of Nbs. (**D**) Example of ELISA results from 24 out of 600 selected clones. Only clones with signal above the red-dashed line were selected for further characterization. (**E**) The DNA of those clones was sequenced and analyzed by similarity. A phylogenetic tree made with those clones shows that they distribute into three groups. Thick lines mark the branch encoding Nbs with high reactivity and specificity. Nb C1 was chosen for further studies.

The online version of this article includes the following figure supplement(s) for figure 1:

**Figure supplement 1.** Construction of phage library.

**Figure supplement 2.** Amino acid sequence of highly reactive nanobodies to hASIC1a.

Subsequently, Nb.C1 was added to large-scale preparations of crude membranes from 293 F cells expressing an affinity-tagged construct (functional hASIC1a comprising amino acids 12–478 with a Strep Tag II in the N-terminus) prior to solubilization with DDM detergent. After affinity purification, the hASIC1a-Nb.C1 complex was pure and monodispersed (*Figure 2A*). This contrasts with the protein aggregates observed in size exclusion chromatography (SEC) of solubilized hASIC1a alone (*Figure 2—figure supplement 1*). In the complex, proteins from the peak of the SEC separated by gel electrophoresis showed bands for hASIC1a and Nb.C1 consistent with a stoichiometric ratio of 1:1 (*Figure 2B*). The solubilized hASIC1a-Nb.C1 complex at 3.8 mg/mL was used to make cryo-EM samples. Images were obtained on a Krios microscope (Thermo Fisher Scientific) using an energy filter and electron-counting camera. Some 2D classes of picked particles are shown in *Figure 2C*, and further processing described in *Figure 2—figure supplement 2* and *Table 1* led to a map of hASIC1a-Nb.C1 complex (*Figure 2D*).

We determined the cryo-EM structure of hASIC1a-Nb.C1 complex at pH 7.4 with estimated resolution of 2.9 Å (*Figure 3*; *Figure 3—figure supplement 1*; *Table 1*). In agreement with other ASIC structures, hASIC1a is a trimer with each subunit ECD resembling the architecture of a hand. At pH 7.4 the thumb domain is away from the finger producing an expanded acidic pocket as observed previously in cASIC1 (*Yoder et al., 2018*) and hASIC1a (*Sun et al., 2020*). The TMD shows a closed pore and the domain-swapped TM2 helixes that define the GAS belt of the pore. We also observed protein densities in the EM map inside the lower pore that were weak, presumably from disorder. When the map was filtered to 7 Å resolution (*Figure 3—figure supplement 2*) density for a reentrant loop with two short helices was observed, consistent with the loop described by *Yoder and Gouaux, 2020*. In cASIC1 the reentrant loops, one from each subunit, form the lower ion permeation pathway. In each loop, the short linker between the helices Re-1 and Re-2 contains

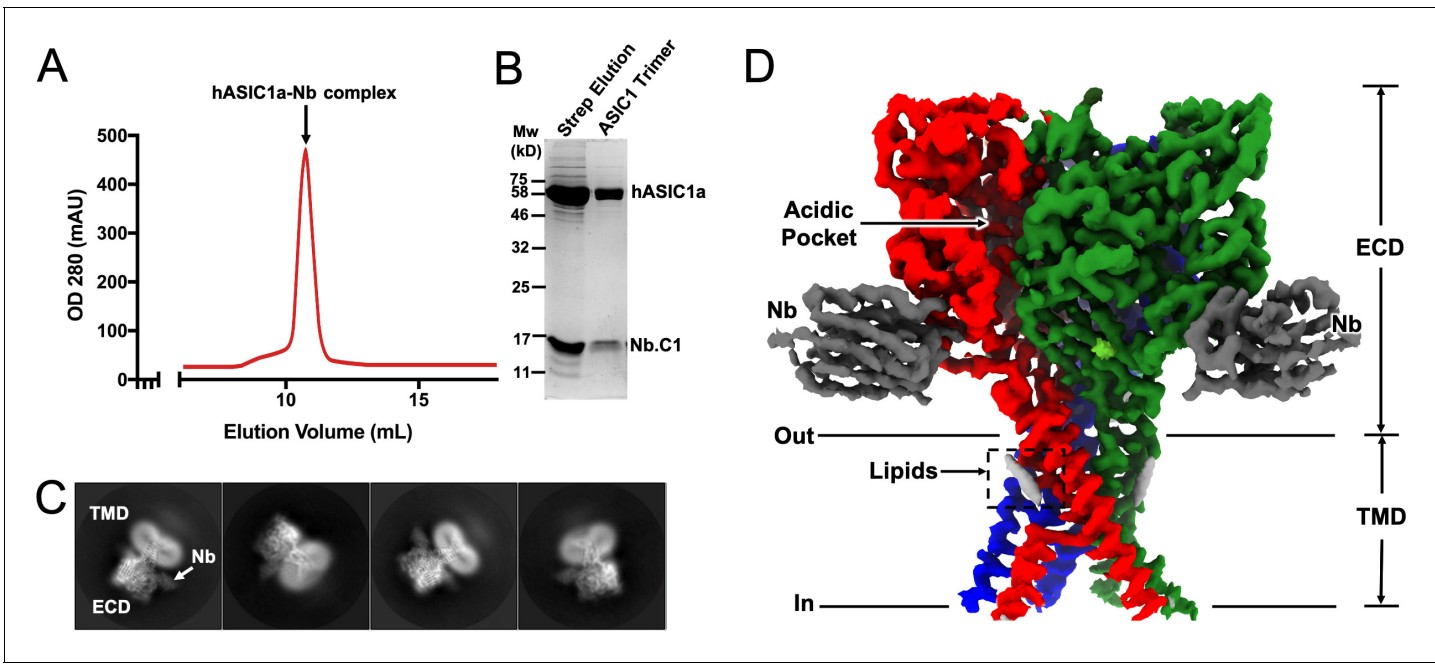

**Figure 2.** Structural determination of human ASIC1a in complex with Nb.C1. (A) Size exclusion chromatography (SEC) purification of the hASIC1a-Nb. C1 complex elutes as a single monodispersed peak. (B) Coomassie blue-stained SDS-PAGE shows two bands corresponding to the molecular weights of hASIC1a and Nb.C1, indicating stable association of the complex that persists after SEC. (C) Representative 2D classes of hASIC1a-Nb.C1 complex particles show distribution in various orientations. The extracellular domain (ECD) and transmembrane domain (TMD) can be readily distinguished as well as Nb.C1 attached to the ECD. (D) Representative view of the 3D density map shows the Nb.C1 in complex with hASIC1a. The three hASIC1a subunits are shown in green, red, and blue; Nb.C1s are shown in dark gray. Lipids are seen attached to the TMD (light gray).

The online version of this article includes the following figure supplement(s) for figure 2:

**Figure supplement 1.** Representative examples of size exclusion chromatography (SEC) profiles of hASIC1a purified in 1% dodecylmaltoside (DDM) or 1% Fos-choline14 in the absence of nanobody.

**Figure supplement 2.** Cryo-electron microscopy (cryo-EM) data processing pipeline for hASIC1a-Nb complex at pH 7.4.

Table 1. Cryo-electron microscopy (cryo-EM) data collection, refinement, and validation statistics.

| Data collection and processing | hASIC1a-Nb |
| --- | --- |
| Magnification | 105,000 |
| Voltage (kV) | 300 |
| Electron exposure (e/Å$^2$) | 45.3 |
| Defocus range (μm) | −1.0 to −2.0 |
| Pixel size (Å) | 0.83 |
| Symmetry imposed | C3 |
| Initial particle images (no.) | 1,287,029 |
| Final particle images (no.) | 84,500 |
| Map resolution (Å) | 2.86 |
| FSC threshold | 0.143 |
| Refinement | |
| Initial model used (PDB code) | 6VTL |
| Model resolution (Å) | 3.7 |
| FSC threshold | 0.5 |
| Map sharpening B factor (Å) | −15 |
| Model composition | |
| Non-hydrogen atoms | 4000 |
| Protein residues | 540 |
| Ligands | 2 |
| Bonds (RMSD) | |
| Length (Å) (# > 4σ) | 0.012 |
| Angles (°) (# > 4σ) | 0.93 |
| MolProbity score | 1.69 |
| Clash score | 6.87 |
| Ramachandran plot (%) | Ramachandran plot (%) |
| Outliers | 0.00 |
| Allowed | 3.17 |
| Favored | 96.83 |
| Rama-Z (Ramachandran plot Z-score, RMSD) | Rama-Z (Ramachandran plot Z-score, RMSD) |
| Whole (N = 536) | 1.31 (0.36) |
| Helix (N = 136) | 0.35 (0.44) |
| Sheet (N = 103) | 1.59 (0.45) |
| Loop (N = 297) | 0.77 (0.37) |
| Rotamer outliers (%) | 0.00 |
| Cβ outliers (%) | 0.00 |

the highly conserved HG motif (His28, Gly29) that, in combination with the GAS belt, is involved in channel gating and selectivity.

Three Nbs are observed bound to the trimeric channel, each in contact with a subunit at the end of the α4-helix and the extended loop of the thumb domain (*Figure 3A*). The most noticeable structural difference between hASIC1a and cASIC1 is in this long loop that extends down from the α4-helix to the tip of the thumb (*Figure 3B*). The loop is longer and more twisted in hASIC1a because it has two extra amino acids D298 and L299 (DL) that are absent in cASIC1 and in all other known ASIC isoforms. These two amino acids are essential for binding of Nb.C1 as their deletion from hASIC1a eliminates the signal in immunofluorescence microscopy. Channels that do not encode

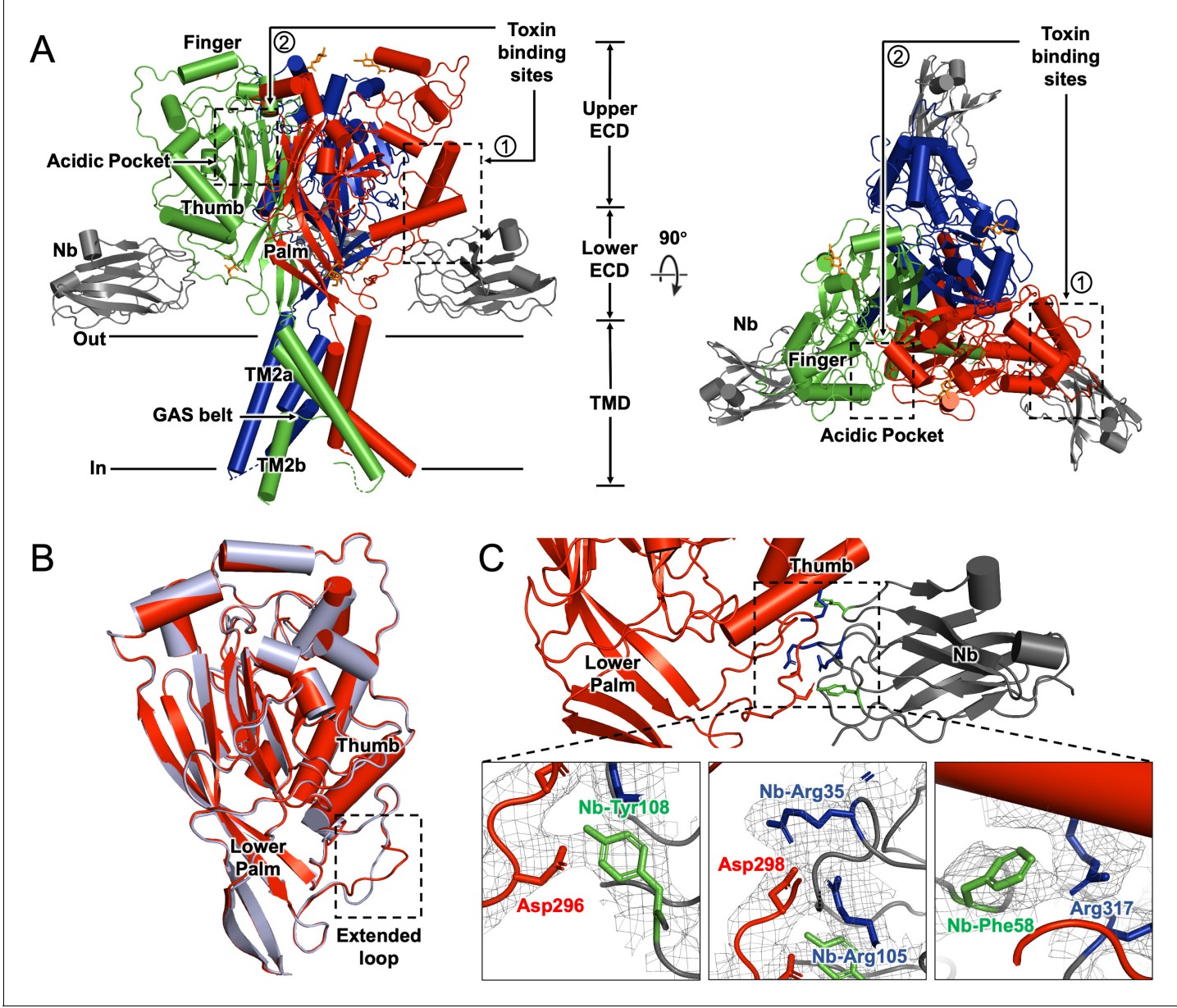

**Figure 3.** Cryo-electron microscopy (cryo-EM) structure of hASIC1a-Nb.C1 complex in the closed conformation. (**A**) Overall structure of hASIC1a-Nb complex in side and top views. Trimeric hASIC1a subunits are shown in red, green, and blue. Nanobodies Nb.C1 attached to the thumb domain of each hASIC1a subunit are shown in dark gray. General location of the overlapping binding sites of MitTx and Mambalgin-1 is indicated by the dashed rectangle (1) while the binding site of PcTx-1 is indicated by dashed rectangle (2). (**B**) Superposition of hASIC1a secondary structure (red) with cASIC1 (6vtl) (light blue) shows substantial differences only in the extended loop of thumb domain. (**C**) Detailed interactions between hASIC1a and Nb.C1 are Asp296-Tyr108 (lower left panel), Asp298-Arg35 and Arg105 (lower middle panel), Arg317-Phe58 (lower right panel). Map densities shown as a mesh. The negatively charged residues are in red, positively charged residues in blue, and aromatic residues are in green.

The online version of this article includes the following figure supplement(s) for figure 3:

**Figure supplement 1.** Cryo-electron microscopy (cryo-EM) imaging of hASIC1a-Nb complex at pH 7.4 and 4 mM $Ca^{2+}$.

**Figure supplement 2.** Intracellular densities in the hASIC1a-Nb electron microscopy (EM) map.

**Figure supplement 3.** Immunoreactivity of HA-tagged Nb.C1 with various species and isoforms of ASIC.

**Figure supplement 4.** Structural comparison of hASIC1-Nb.C1 complex with hASIC1 at high pH.

**Figure supplement 5.** Functional characterization of hASIC1a-Nb.C1 and channels with or without DL residues.

those two residues such as cASIC1, mASIC1a, mASIC2a, and mASIC3 are also not recognized by Nb.C1 in IF experiments (*Figure 3—figure supplement 3*). The residues involved in interactions with Nb.C1 are shown in *Figure 3C*. Two Asp residues in the extended loop of the thumb are involved: D296 interacts with Nb-Y108, and D298 with Nb-R35 and Nb R105. In addition, R317 at the end of α4 interacts with Nb-F58. None of these interactions would be possible in cASIC1. The loop in the hASIC1a structure of *Sun et al., 2020* is poorly resolved, but in the current structure is well defined likely because of stabilization provided by Nb.C1, as shown in *Figure 3—figure supplement 4*.

Of note, extensive functional analysis of hASIC1a bound to Nb.C1 showed that Nb binding did not change the channel's properties. The average magnitude of current, rate of desensitization, midpoint pH of activation ($pH_{50a}$), and of steady-state desensitization ($pH_{50ssd}$) were all unchanged (*Figure 3—figure supplement 5A–B*). The presence of the DL motif itself, however, affects the pH dependence of activation and desensitization. DL lies within a five-amino-acid stretch of hASIC1a which, when mutated, confers the altered pH sensitivity of the mouse isoform on hASIC1a (*Sherwood and Askwith, 2008*). Further, we compared the effects of the DL motif on hASIC1a and cASIC1. Deletion of DL in hASIC1a produces a small but significant left shift of the $pH_{50ssd}$ (from 7.11±0.02 to 7.18±0.02), while insertion of DL into cASIC1 produces the opposite effect (from 7.45±0.01 to 7.38±0.01) (*Figure 3—figure supplement 5C–D*).

To visualize the binding sites of MitTx, PcTx1, and Mambalgin-1 and compare them to the binding site of Nb.C1, our structure of the hASIC1a-Nb.C1 complex was superimposed on chicken or human subunits bound to each of the three toxins, and shown in orthogonal views (*Figure 4*). The superimposition of cASIC1-MitTx1 in the open conformation shows the α-subunit of the toxin interacting with the tip of the thumb close to the membrane bilayer, and with the extended chain that connects α5 to the β10 strand. Meanwhile, the β-subunit forms contacts with α4 and α5 of the thumb, extensively overlapping with the binding site of PcTx1 but not with that of Nb.C1. Though the binding sites of Nb.C1 and MitTx subunits are different, the bulky scaffold of the Nb produces steric hindrance to binding of the α-subunit, marked in the figure by a dashed square. In contrast, Nb.C1 and PcTx1 bind at distinct and well-separated sites, preventing mutual interference when both peptides bind simultaneously to the surface of the channel (*Figure 4B*).

Mambalgin-1 and Nb.C1 share common interactions: residue D298 of hASIC1 interacts with K8 in Mambalgin-1, and hASIC1 residues at the end of α4 interact with both polypeptides. The dashed square indicates the area of clashes between Nb.C1 and toxin (*Figure 4C*). Although we did not test here the effect of Nb.C1 on Malganbin-1 binding, the results predict a decrease in the functional effects mediated by Malganbin-1.

## Nb.C1 antagonizes binding of MitTx to hASIC1a

The predicted overlap of Nb.C1 with the MitTx α-subunit binding site raises the possibility that Nb.C1 could interfere with the action of MitTx on hASIC1a. To that end, oocytes expressing hASIC1a were incubated with 50 nM of Nb.C1 for 15 min; they were then exposed to 50 nM of MitTx at pH 7.4 while measuring currents with two-electrode voltage clamp. MitTx-induced currents were decreased in oocytes pretreated with the Nb (*Figure 5A–D*), consistent with the Nb.C1 interfering with toxin binding to hASIC1a.

## Increased potency of PcTx1 by tethering to Nb.C1

PcTx1 functions as an inhibitor of ASIC1a by shifting the steady-state desensitization toward more alkaline pH (*Chen et al., 2005*; *Saez et al., 2011*). Two crystal structures of cASIC1 with PcTx1 show that the toxin binds to the α5-helix of the thumb, at the interface with the palm domain of the adjacent subunit (*Dawson et al., 2012*; *Baconguis and Gouaux, 2012*). The apparent $EC_{50}$ of PcTx1 for hASIC1a has been reported in the range 0.9–3.7 nM and the time constant of recovery from inhibition is reported as 125 s (*Chen et al., 2021*) or 87 s (*Chassagnon et al., 2017*). We hypothesized that slowing the recovery from inhibition would be advantageous for in vivo applications of PcTx1 either as a potential pain suppressor or for protecting neurons from ischemia of the brain. That aim could be achieved by fusing Nb.C1 to PcTx1 in a single polypeptide provided that the Nb exhibits slower dissociation than the toxin and that the fusion does not interfere with binding to hASIC1a. To verify binding we first used SH-SY5Y cells, which are derived from a human neuroblastoma and

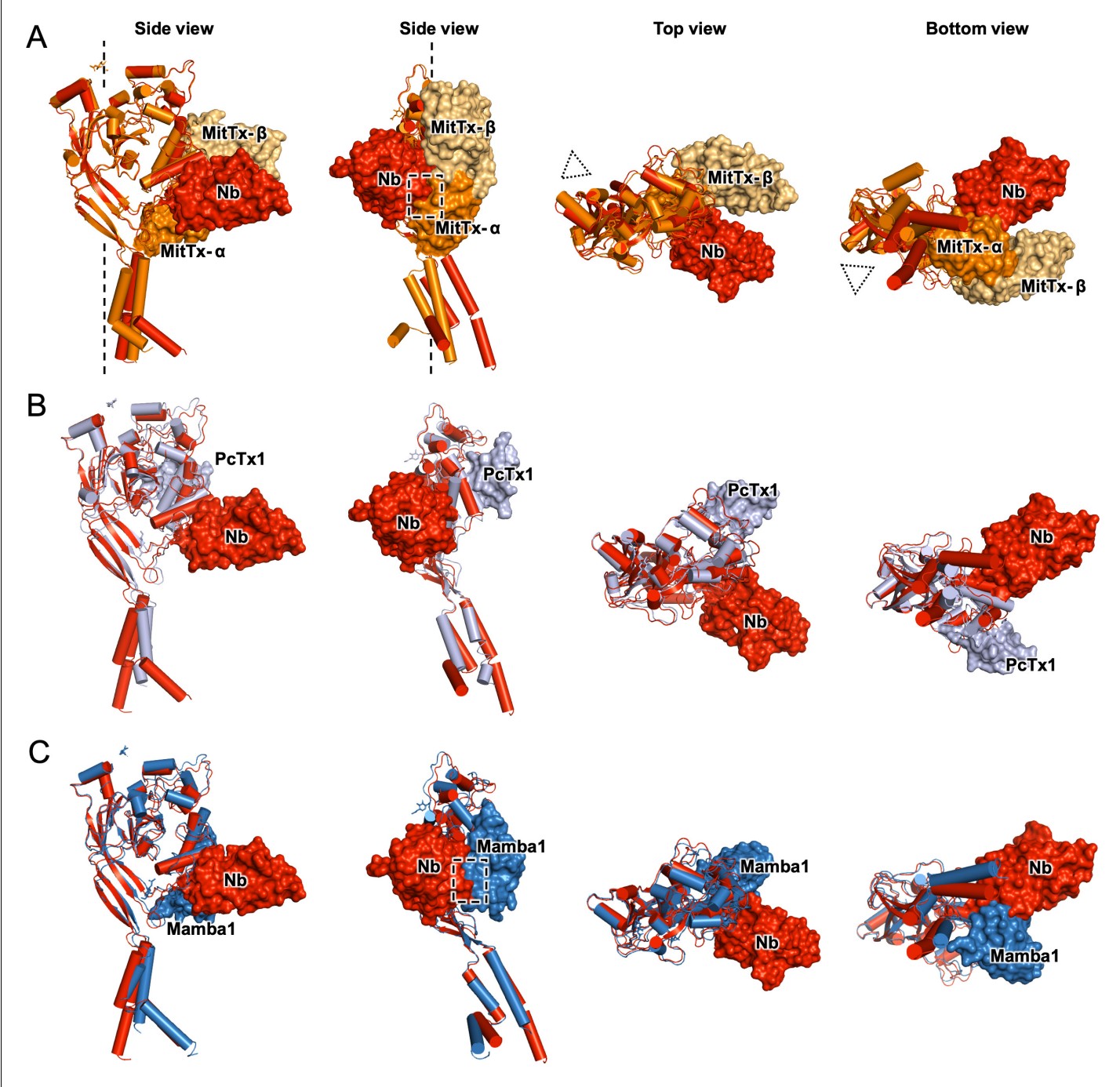

**Figure 4.** Structural comparison of hASIC1a-Nb.C1 complex to toxin-bound ASICs. Two side, top and bottom views of superimposed structures of hASIC1a-NbC1 complex (red) with (**A**) MitTx-bound to chicken ASIC1 (4ntw) in open conformation (orange). In side views, the threefold axis of the channel is indicated by a dashed vertical line; in top and bottom views it is indicated by dotted triangles. (**B**) PcTx1-bound chicken ASIC1 (3s3x) (gray). (**C**) Mambalgin-1-bound human ASIC1 (7ctf) (blue). Only one subunit is shown for simplicity. Surface clashes are indicated by dashed rectangles. Nb.C1, MitTx- α, MitTx- β, PcTx1, Mambalgin-1 are shown as red, orange, light-orange, light-purple, marine respectively.

express endogenous hASIC1a (*Xiong et al., 2012*). An Nb.C1-PcTx1 fusion construct produced a fluorescence signal that decorated the plasma membrane of SH-SY5Y (*Figure 5E–G*).

The time course of dissociation of Nb.C1 from hASIC1a was followed in transfected HEK293 cells using the following protocol. Live cells were incubated with 10 nM Nb.C1 for 30 min at 18°C. After several washes to remove the unbound Nb.C1, cells were followed for four intervals of 1 hr duration.

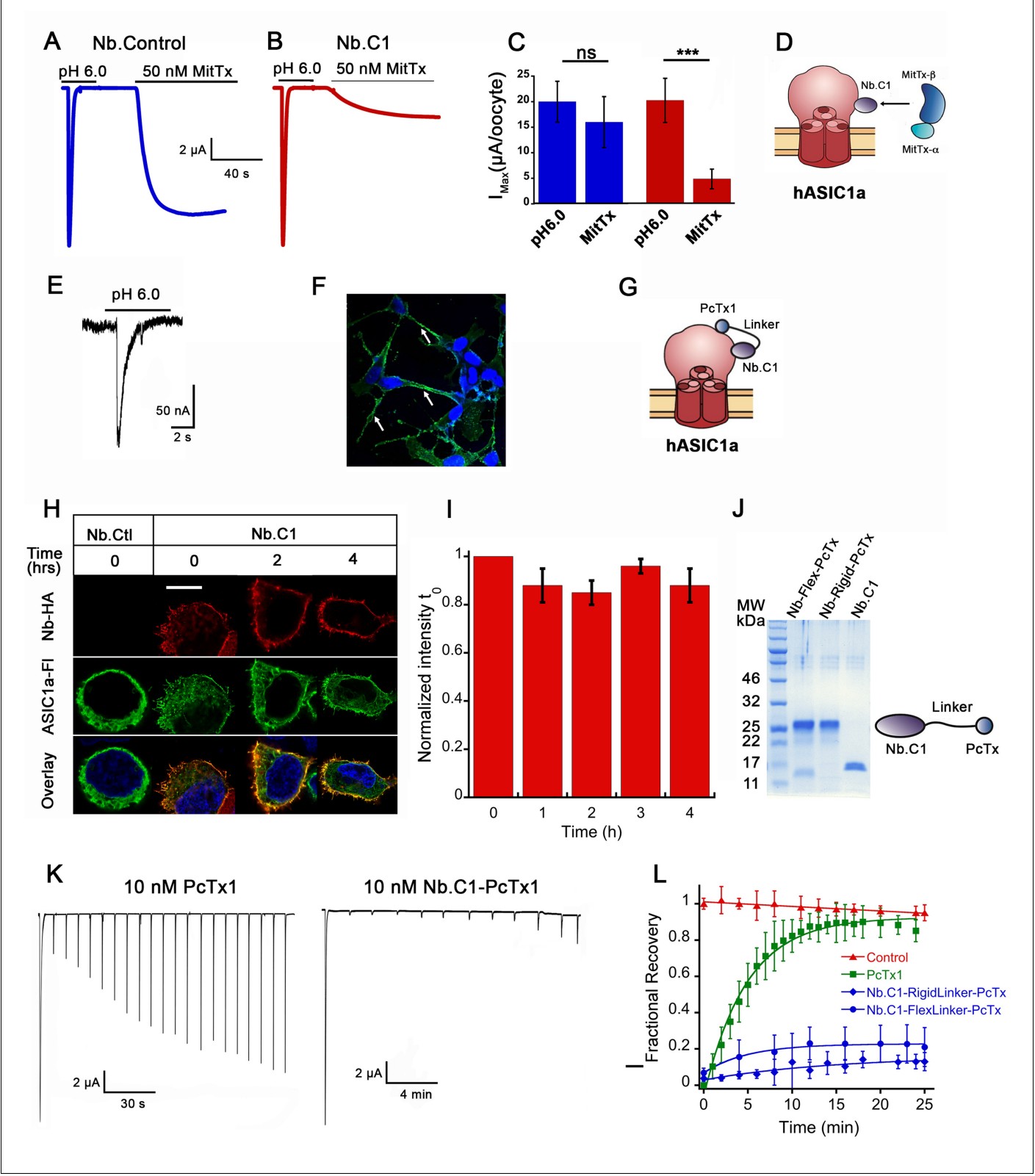

**Figure 5.** Effects of Nb.C1 on MitTx and PcTx1 binding to hASIC1a. (A) Representative currents of an oocyte expressing hASIC1a activated with pH 6.0 followed by a second activation with 50 nM MitTx at pH 7.4. (B) Same experiment after pre-incubation of the oocyte with 50 nM Nb.C1 for 15 min. (C) Summary of the peak currents from pH 6.0 and MitTx activations. In this and all traces, the conditioning pH is 7.4. The bars represent the mean±SD of currents, n=8 Nb control and n=6 Nb.C1. Asterisks indicate statistical significance in t-test, p<0.001. (D) Cartoon of the proposed mechanism of how

*Figure 5 continued on next page*

*Figure 5 continued*

Nb.C1 associated with hASIC1a may interfere with MitTx binding. (E) Whole-cell patch clamp of SH-SY5Y cells activated with pH 6.0 generates typical hASIC1a currents. Proton-induced currents are inhibited by PcTx and amiloride. (F) Immunofluorescence confocal image of SH-SY5Y cells incubated with Nb.C1-PcTx1-HA fusion and anti-HA antibody (green) shows cells decorated on the periphery. Nuclei were stained with DAPI (blue). Scale bar, 5 μm. (G) Cartoon representation showing the Nb.C1-PcTx1 polypeptide binding to two distinct sites on the surface of hASIC1a, accounting for a possible mechanism of toxin potentiation. (H) Confocal images of live HEK-293 cells transfected with hASCIC1a-Flag on coverslips incubated with Nb.C1-HA for 30 min and followed for 0, 1, 2, 3, and 4 hr at 18°C in DMEM containing HEPES. Three of the five time points are shown. At each 1 hr interval, all cells were washed except for the one dish of cells removed for fixation. All cells were processed for immunofluorescence with HA and Flag monoclonals to visualize Nb.C1-HA and hASIC1a-Flag, respectively. Nb.C1-HA labels only the cell surface whereas hASIC1a distributes in the plasma membrane and intracellular endoplasmic reticulum and perinuclear membrane. Scale bar, 5 μm. (I) Quantification of fluorescence intensity of Nb.C1 (red channel) normalized to time 0 hr ($t_0$). For each time point 300 cells were analyzed. Columns are the mean ± SEM. (J) Coomassie blue SDS-PAGE of purified fusion proteins (Nb.C1-FlexLinker-PcTx and Nb.C1-RigidLinker-PcTx) and Nb.C1 alone. On the right a cartoon representation of the fusion proteins. (K) Representative examples of oocytes expressing hASIC1a exposed to 10 nM of PcTx1 or 10 mM of Nb.C1-Rigid-PcTx1 fusion for 60 s prior to serial activations with a change of pH from 7.35 to 6.0. Cells remained in the perfusion chamber throughout the experiment. (L) Time course of recovery of acid-induced currents in control (no pretreatment), and pretreatment with PcTx1, Nb.C1-Flex-PcTx, or Nb.C1-Rigid-PcTx1. Preconditioning pH 7.35, activation pH 6.0. Data were fit with a single exponential $a\left(1 - e^{-t/\tau}\right)$ where $\tau$ is 220 s for PcTx, 350 and 880 s for Nb.C1-Flex-PcTx and Nb.C1-Rigid-PcTx; a = 0.90 for PcTx, and 0.16 and 0.14 for the fusions, respectively. Data points represent the mean ± SD of 7–12 cells. Values of currents from each cell are shown in *Source data 3*.

At each of the four time points, a group of cells was fixed and processed for IF or washed again until the next time point. Confocal images showed no decay of the Nb.C1 signal during the course of the experiment indicating that the Nb.C1 remains bound to the channel with no significant decay through the 4 hr observation period (*Figure 5H–I*).

To estimate the Nb binding affinity, we first measured binding of Nb.1C to immobilized, detergent-solubilized hASIC1a protein using surface plasmon resonance. From experiments with two different Nb and hASIC1a protein preparations, global fits to association time courses with Nb concentrations from 0.4 to 100 nM yielded association rate constants $k_{on}$ ranging from $6.8 \times 10^4$ to $2.2 \times 10^5$ M$^{-1}$ s$^{-1}$. Unfortunately, the dissociation time courses (measured for up to 1 hr) from these experiments were too brief to observe the full dissociation time course, but a rate constant $k_{off} \leq 1.4 \times 10^{-5}$ s$^{-1}$ (0.05 hr$^{-1}$) can be estimated from the live-cell binding data of *Figure 5I*. Taking the smallest $k_{on}$ value as a lower bound on association rate yields the equilibrium constant $k_d \leq 0.2$ nM.

Next, three constructs were made in which PcTx1 was tethered to the carboxyl-terminus of Nb.C1 with either a flexible or rigid 30 a.a. linker, and another with PcTx1 tethered to a control anti-GFP Nb (Nb.Control). His$_6$ and HA tags were added to the C-termini of the three constructs for affinity purification and detection by IF. The proteins were expressed in bacteria, isolated from the periplasm and further purified as indicated in methods (*Figure 5J*).

To determine the effect of the Nb.C1-PcTx1 on the kinetics of PcTx1 inhibition, proton-induced currents in oocytes expressing hASIC1a were examined by TEVC according to the protocol shown in *Figure 5K–L*. After a first activation with pH 6.0, a 60 s incubation with 10 nM PcTx, Nb.C1-Flex-PcTx1, or Nb.C1-Rigid-PcTx was followed by subsequent activations for a total period of 25 min. Oocytes not exposed to any peptide served as control for spontaneous decay of the currents after multiple consecutive stimuli. Cells pre-incubated with PcTx1 recovered current faster ($\tau_{off}$ 220 s) and more completely (90 ± 16%) than those pre-incubated with the fusion proteins (*Figure 5I–J*). Cells exposed to Nb.C1-Flex-PcTx recovered only a small fraction of the initial current after 25 min (0.16) even though the calculated $\tau_{off}$ (355 s) was not much slower than that of toxin alone whereas Nb.C1-Rigid-PcTx reached a lower plateau level (0.14) and slower rate of recovery ($\tau_{off}$ = 882 s) compared to the flexible linker (*Source data 3*). The partial recovery may be attributed to the presence of some free toxin from cleavage of the fusion proteins and was observed most prominently with the flexible linker (*Figure 5J*). Together these experiments indicate that PcTx1 tethered to Nb.C1 significantly potentiates the action of the toxin.

## Discussion

Notwithstanding the extensive progress achieved in the elucidation of the structure and functions of ASICs in recent years, important challenges still remain, underscoring the need to develop new tools

to further our understanding of these channels. To that end, we sought to develop Nbs specific to hASIC1a. This strategy exploits the capacity of the immune system to produce a large variety of small polypeptides against a single target, each with unique properties that can be selected for specific applications.

Here, we show that one Nb (Nb.C1) isolated from a phage display library prevents aggregation and stabilizes hASIC1a, markedly improving the quality of the sample preparation for cryo-EM studies. We obtained the structure of hASIC1-Nb.C1 complex in the closed conformation at 2.9 Å resolution, indicating that the Nb has potential toward attaining high-resolution structural information of hASIC1a in additional conformations. The growing use of Nbs in structural biology stems from the fact that they can stabilize native conformations of proteins, in particular membrane proteins, facilitating structural analysis by crystallization or cryo-EM.

The Nb also constitutes a new tool to accelerate the development of therapeutic agents targeted to hASIC1a. The cryo-EM structure of the hASIC1a-Nb.C1 complex reveals that Nb.C1 binds to the thumb domain of hASIC1a to an epitope in the extended loop of the thumb that contains two extra amino acids (D298 and L299) unique to hASIC1a. These residues determine the high specificity toward hASIC1a and their absence explains the lack of reactivity with other species or isoforms (cASIC1, mASIC1, ASIC2, or ASIC3). Nb.C1 binding also spans to the end of the $\alpha$4-helix. This is a region on the surface of the ECD that is also recognized by various toxin polypeptides with functional activity toward many ASIC isoforms. Specifically, it overlaps with the binding site of Mambalgin-1 (*Sun et al., 2020*) and the binding site of the $\alpha$-subunit of MitTx. This last toxin is the component of the Texas coral snake venom that produces severe pain (*Greene et al., 2020*) by activating hASIC1a in peripheral neurons (*Bohlen et al., 2011*). In these two instances, most of the binding interference with the Nb is due to steric hindrance produced by the scaffold of the Nb, which is large (16.28 kDa) compared to the size of Mambalgin-1 (6.55 kDa) and MitTx $\alpha$-subunit (7 kDa). Therefore, Nb.C1 could serve as a competitive antagonist to MitTx and as a potential antidote for the pain-producing component of the snake bite. Nbs offer practical advantages over currently available antivenoms usually raised in sheep or horses (*Yang et al., 2017*) because Nbs can be produced in bacteria in large amounts and at low cost.

The Nb.C1-binding site is distinct from that of the inhibitory toxin PcTx. The lack of competition for binding sites and steric interference but yet close proximity of the two binding sites offers the possibility of using Nb.C1 as a carrier for PcTx1 to increase potency and decrease off-target effects of the toxin. While the effect of PcTx1 alone is rapidly reversible, a fusion protein incorporating Nb.C1 achieves inhibition of ~84–87% that persists for more than 30 min in functional studies and remains bound for more than 4 hr on the surface of live cells according to IF assays. The changes in toxin kinetics are consistent with the Nb having a much slower $k_{off}$ rate than that of the toxin. Further optimization of the sequence of the linker, mainly to minimize or eliminate cleavage, could increase even more the fractional current inhibited by the Nb.C1-PctX fusion peptide. Tethering of PcTx to the Nb also would direct the toxin to bind preferentially to the subunit ASIC1a increasing specificity, which is beneficial for in vivo applications such as the amelioration of ischemic damage to the brain (*McCarthy et al., 2015*; *Yang et al., 2011*; *Xiong et al., 2004*).

Why does Nb.C1 bind to sites overlapping with those of toxins (*Figure 4*) but itself does not alter channel function? We note that Nb.C1 does not interact with the $\alpha$5-helix of the thumb, while the three toxins in consideration all bind to $\alpha$5 and these interactions change the conformation of $\alpha$5 in a toxin-specific manner. Many lines of evidence support the notion that displacement of $\alpha$5 is an essential component of channel gating. Mutations in $\alpha$5 (*Jasti et al., 2007*; *Vullo et al., 2017*), formation of cysteine bonds that restrain movement of helix (*Yoder et al., 2018*; *Chen et al., 2021*), and binding of polypeptides that produce small but significant displacements are maneuvers that alter gating by lowering $pH_{50a}$, abolishing currents, inducing desensitization or causing channel opening. We posit that the absence of Nb.C1 contacts with $\alpha$5 is the most likely explanation of why Nb.C1 does not change function of hASIC1a. Interactions of Nb.C1 are limited to a short segment of the extended loop of the thumb that projects down to TM1 and a segment of the $\alpha$4-helix of the thumb. Comparison of our structure with the hASIC1a structure of *Sun et al., 2020* shows that Nb.C1 binding produces a very small movement of $\alpha$4 and negligible perturbation of $\alpha$5 (*Figure 3—figure supplement 4*).

In summary, we show examples of uses of Nb.C1 isolated from an hASIC1a-specific phage display library to advance structural and functional studies of the human channel, and as means to increase or attenuate effects of ASIC-specific toxins with potential therapeutic applications.

# Materials and methods

### Key resources table

| Reagent type (species) or resource | Designation | Source or reference | Identifiers | Additional information |
|---|---|---|---|---|
| Gene (*Homo sapiens*) | ASIC1a | GenBank | NCBI Ref Seq: NP_001086.2 | |
| Strain, strain background (*Escherichia coli*) | TG1 | Lucigen | Cat#: 60502 | Electrocompetent cells |
| Strain, strain background (*Escherichia coli*) | WK6 | ATCC 47078 | Thermo Fisher Scientific | Expression of nanobody proteins |
| Strain, strain background (*Escherichia coli*) | DH5α | Max efficiency DH5α | Cat#: 18258012 | Electrocompetent cells |
| Cell line (*Homo sapiens*) | HEK293T | ATCC 47078 | ATCC 47078 | |
| Cell line (*Homo sapiens*) | FreeStyle 293 F cells | Thermo Fisher Scientific | Cat#: R79007 | |
| Cell line (*Homo sapiens*) | SH-SY5Y | ATTC | ATTC CRL-2266 | |
| Recombinant DNA reagent | pADL-22c | Antibody Design Labs | Cat#: PD0110 | Phagemid for construction of nanobody library |
| Recombinant DNA reagent | CM13 Helper phage | Antibody Design Labs | Cat#: PH020L | Rescue phagemid library |
| Recombinant DNA reagent | pcDNA3.1 | Invitrogen | Cat#: V790-20 | Vector |
| Antibody | Goat anti-llama polyclonal antibody HRP | NOVUS | Cat#: NB7242 | Detection of anti-ASIC1a antibodies in alpaca serum (1/1000) |
| Antibody | Anti-HA rabbit monoclonal | Cell Signaling C29F4 | Cat#: 3724T | IF (1/1000) |
| Antibody | Anti-Flag mouse monoclonal M2 | Sigma-Aldrich | Cat#: F1804 | IF (1/1000) |
| Antibody | Anti-M13 g8p antibody HRP mouse monoclonal | Antibody Design Labs | Cat#: AS003-100 | For phage ELISA (1/5000) |
| Peptide, recombinant protein | PcTx1 | Alome | Cat#: STP-200 | |
| Peptide, recombinant protein | Alpha/beta MitTx | Alome | Cat#: M-100 | |
| Polypeptide, recombinant proteins | Alpaca nanobodies | This study | | Isolated from phage display library of immunized alpaca with hASIC1a |
| Commercial assay or kit | QuickChange mutagenesis | Agilent Technologies | Cat#: 200521 | Mutagenesis of DNA |
| Commercial assay or kit | ProtoScrript II First strand cDNA | New England Biolabs | Cat#: E6560L | Synthesis of single strand DNA |

*Continued on next page*

*Continued*

| Reagent type (species) or resource | Designation | Source or reference | Identifiers | Additional information |
|---|---|---|---|---|
| Chemical compound, drug | Pierce anti-HA magnetic beads | Thermo Fisher Scientific | Cat#: 88837 | Affinity purification of HA-tag proteins |
| Chemical compound, drug | Monoclonal Anti-HA agarose | Sigma-Aldrich | Cat#: A2095 | Affinity purification of HA-tag proteins |
| Chemical compound, drug | Strep Tactin Resin | IBA | Cat#: 2-1201-002 | Affinity purification of Strep-tag proteins |
| Chemical compound, drug | Ni-NTA Agarose | Qiagen | Cat#: 30210 | Affinity purification of nanobodies from periplasm |
| Chemical compound, drug | Cholesterol Hemisuccinate tris | Anatrace | Cat#: CH210 | |
| Software, algorithm | MotionCor2 | DOI: 10.1038/nmeth.4193 | RRID:SCR_016499 | http://msg.ucsf.edu/em/software/motioncor2.html |
| Software, algorithm | Gctf | DOI: 10.1016/j.jsb.2015.11.003 | RRID:SCR_016500 | https://www.mrc-lmb.cam.ac.uk/kzhang/Gctf/ |
| Software, algorithm | RELION 3.1 | DOI: 10.7554/eLife.42166 | RRID:SCR_016274 | http://www2.mrclmb.cam.ac.uk/relion; |
| Software, algorithm | PHENIX | | RRID:SCR_014224 | https://www.phenixonline.org; |
| Software, algorithm | Coot | DOI: 10.1107/S0907444910007493 DOI: 10.1107/S0907444910007493 | RRID:SCR_014222 | https://www2.mrc-lmb.cam.ac.uk/personal/pemsley/coot/ |
| Software, algorithm | MolProbity | DOI: 10.1107/S0907444909042073 | RRID:SCR_014226 | RRID:SCR_014226 |
| Software, algorithm | Pymol | PyMOL Molecular Graphics System, Schrodinger, LLC | RRID:SCR_000305 | RRID:SCR_000305 |
| Software, algorithm | UCSF Chimera | DOI: 10.1002/jcc.20084 | RRID:SCR_004097 | http://plato.cgl.ucsf.edu/chimera/ |
| Software, algorithm | UCSF ChimeraX | DOI: 10.1002/pro.3235 | RRID:SCR_015872 | http://cgl.ucsf.edu/chimerax/ |
| Software, algorithm | CCP-EM | DOI: 10.1002/pro.3235 | | https://www.ccpem.ac.uk/ |
| Software, algorithm | DemoPIcker | This study | | https://github.com/fsigworth/aEMCodeRepository/tree/master/Teaching/PartPickingDemo, (*Sigworth, 2021*; copy archived at swh:1:dir:2cdf6a8a6b19d8be1408954f51bf9d81e44edb11) |
| Other | Series S Sensor Chip CM5 | Cytiva | Cat#: 29104988 | For Biacore (GE) instrument |

## Alpaca immunization

A male alpaca was immunized with intact HEK293T cells (ATCC CRL-11268) transfected with hASIC1a according to the schedule shown in *Figure 1—figure supplement 1*. Antigen expressing cells and adjuvant were injected subcutaneously in adjacent sites to preserve the native conformation of hASIC1a. Blood was obtained before the first injection, and 1 week after the third and fifth injections. *Figure 1* depicts the subsequent steps of construction and screening of the phage display library. Immunization and bleeding of alpaca were conducted with the assistance of a veterinarian, and protocols were approved by IACUC of Tsinghua University (protocol number 07749). The

Association for Assessment and Accreditation of Laboratory Animal Care International (AALAC) has accredited Tsinghua University veterinarians and facilities.

## Isolation of alpaca IgGs

One-hundred microliter of protein G agarose beads (Sigma) were added to 0.5 mL of alpaca serum and incubated for 1 hr followed by three washes with phosphate buffer saline (PBS). The IgG3 fraction bound to the beads was eluted with 0.5 mL of 150 mM NaCl, 0.58% acetic acid, pH 3.5, and immediately neutralized with 1 M Tris-HCl pH 8.0. Beads were washed and the IgG1 fraction still bound to the G agarose was eluted with lower pH: 0.5 mL of 100 mM glycine-HCl pH 2.7 followed by neutralization. The initial flow through from protein G beads was incubated with 50 µL protein A beads for 1 hr. After washes with PBS, the IgG2 fraction was eluted with 0.5 mL of 150 mM NaCl, 0.58% acetic acid pH 4.5 and neutralized to pH 8.0. Alpaca IgG2/3 represent the single-domain antibodies (*Maass et al., 2007*).

## Construction of phage displayed Nb library

Five days after the final immunization, peripheral blood lymphocytes were isolated from 100 mL of whole blood using Accuspin-Histopaque System (Sigma). Total RNA was extracted with Trizol (Invitrogen); 30 µg of total RNA were used for synthesis of single strand DNA primed with oligo-dT and using Superscript III kit (Life Technologies). The variable region of the heavy chain from IgG2/IgG3 (VHH) domains was amplified by nested PCR. The first PCR was conducted with a pair of primers specific for alpaca annealing to the leader sequence and to IgG CH2 domain. The amplified IgG2/3 dsDNA was gel extracted from the first PCR product by cutting the 700 bp band. The second PCR (18 cycles) was conducted with a pair of primers specific to alpaca IgG FR1 region and IgG2/3 hinge region (*Figure 1—figure supplement 1B*). SfiI restriction sites, generating different sticky ends, were introduced by PCR for cloning into a pADL-22c phagemid vector (Antibody Design Labs). A total of 30 ligation reactions were pooled and electroporated into TG1 competent *Escherichia coli* (Lucigen); 18 clones were randomly picked to examine the efficacy of dsDNA insertion (*Figure 1—figure supplement 1D*). A library of $1\times10^9$ individual transformants was superinfected with CM13 helper phage (Antibody Design Labs) after TG1 F-pilus induction. VHH-domain-displaying bacteriophages were produced overnight by shaking the bacterial culture at 37°C supplemented with IPTG to induce expression of Nb fragments. Phages were isolated from the medium of an overnight culture by two successive precipitation steps with 4% PEG-8000 in 500 mM NaCl. Phages displaying Nb were dissolved in 1 mL PBS followed by selection for binding to hASIC1a.

## Library screening with a customized panning protocol

Panning was conducted using a modified multi-antigen presenting system that maximizes capturing high-affinity native epitope binders. For the first round of panning, *Xenopus* oocytes expressing high levels of hASIC1a at the plasma membrane were used for selection. Cells were pre-incubated with blocking solution (100 mM NaCl, 3 mM KCl, 2 mM CaCl₂, 10 mM HEPES pH 7.5, and 2% skimmed milk) and incubated with phage in the same solution without milk. For the second and third round of panning, affinity-purified hASIC1a bound to either Strep Tactin XT Agarose beads or magnetic beads (IBA) were used for selection. Incubation and washing buffer (150 mM NaCl, 2 mL CaCl₂, 10 mM HEPES pH 7.4, 0.04% DDM ± 2% milk). After three rounds of panning, binding phages were pooled and a sub-library from the third panning was transformed into WK6 –a strain of *E. coli* for Nb expression. A sample of single clones was examined by extraction of crude periplasm proteins and tested by ELISA. Positive clones were sequenced followed by Nb purification.

## ELISA

Affinity-purified hASIC1a protein (5 µg/mL) was coated onto 96-well microtiter plates overnight at 4°C and blocked with 5% skimmed milk. For alpaca IgGs ELISA, different concentrations of IgG1, IgG2, and IgG3 were added and incubated at room temperature (RT) for 2 hr. After three washes, goat anti-llama IgG conjugated with HRP (NOVUS) was added and enzyme reaction was detected with peroxidase substrate ABTS and quantified at 405 nm in a microplate reader. Non-coated or bovine serum albumin-coated wells served as controls. For phage ELISA, hASIC1a-coated plates were incubated with the generated phage library as well as the panned phage sub-libraries in

various phage particle concentrations. Goat anti-M13 monoclonal antibody HRP-conjugated (GE Healthcare) was used for detecting Nb enrichment after consecutive rounds of panning. For periplasmic extract ELISA, single-clone periplasmic extracts from the third panned sub-library were prepared. In brief, individual bacterial clones were cultured in deep-square-96-well plates (Corning) and grown to exponential phase. Nb production was induced by IPTG and the incubation continued overnight at 28°C. Plates were centrifuged and bacterial pellets were resuspended in 100 µL of TES buffer (in mM): 200 Tris-HCl pH 8.0, 0.5 EDTA, 500 sucrose on a vibrating platform at 2000 rpm for 1 hr; 100 µL of ddH$_2$O was added to each well and returned to vibrating platform for 1 hr. Plates were centrifuged and the supernatants – containing Nbs – were recovered and examined by ELISA. Single-clone periplasmic extracts were added to coated ELISA plates. Mouse anti-HA monoclonal antibody (Santa Cruz) and goat anti-mouse-HRP were added to recognize and detect the bound Nbs. Non-coated wells or an irrelevant Nb were used as controls.

## Nb purification

The phagemid vector pADL-22c encoding a bacterial periplasmic secretion leader sequence and amber stop codon was used for Nb purification. A 30 mL LB overnight culture was diluted 1:1000 into 800 mL fresh Terrific Broth in the presence of 100 µg/mL ampicillin. When the culture reached an OD$_{600}$ of 0.8, Nb expression was induced by 1 mM IPTG and the temperature was decreased to 28°C for 16 hr. Bacteria were harvested by centrifugation and the pellet was resuspended in 20 mL of TES buffer. Periplasmic protein extraction was conducted by osmotic shock as indicated above. After centrifugation, the Nbs were affinity-purified with Ni-NTA agarose beads (Qiagen) and eluted with imidazole (*Pardon et al., 2014*).

## hASIC1a expression and purification

Fully functional hASIC1a comprising amino acids 12–478 was tagged in the N-terminus with StrepII tag. The construct was expressed in HEK293F cells (Invitrogen) cultured in suspension. Expression level of hASIC1a was estimated 48–72 hr post-transfection by immunofluorescence using StrepII tag monoclonal (Abcam). One liter of cell culture was routinely used for isolation of hASIC1a. Crude membrane pellets were resuspended in lysis buffer (50 mM HEPES pH 7.4, 150 mM NaCl, 5 mM CaCl$_2$). For hASIC1a-Nb complex purification, crude membranes were first incubated with 3–4 mg of purified Nbs, protease inhibitor cocktail (Roche) under slow agitation at 4°C for 3 hr to allow binding of the Nb. The suspension was then treated with 1% DDM (Anatrace). After clarifying the homogenate by ultracentrifugation, hASIC1a was affinity-purified with Strep Tactin XT agarose beads (IBA) to achieve high degree of purity and to remove free Nbs. Elution was conducted with the addition of 50 mM D-biotin, samples were concentrated to a volume of 0.8 mL using a 50 kDa cutoff Centricon (Millipore). Samples were injected to a size exclusion column (Superdex 200 Increase 10/200 GL, GE Healthcare). Fractions containing the hASIC1a-Nb.C1 micelle complex were pooled and concentrated to 3.8 mg/mL.

## Cryo-EM specimen preparation, data acquisition, and processing of hASIC1a-Nb complex

Quantifoil holey carbon grids (R1.2/1.3 300 mesh Au) were glow-discharged with carbon side facing up for 1 min at 15 mA. Human ASIC1a-Nb affinity-purified protein at pH 7.4 was subjected to SEC (running buffer 20 mM HEPES pH 7.4, 150 mM NaCl, 5 mM CaCl$_2$) and immediately concentrated to 3.8 mg/mL for grid preparation. A 3 µL droplet of sample was applied to the carbon side of each grid. Grids were blotted and plunge-frozen using a Vitrobot apparatus (Thermo Fisher Scientific) with the chamber at 18°C and 100% humidity. In total, 9039 micrographs were collected on Titan Krios microscopes (Thermo Fisher Scientific) operated at 300 keV. Images were collected using SerialEM (*Schorb et al., 2019*) with an image shift pattern of 3× three holes, with one shot per hole. The detector was a Gatan K3 camera positioned after an energy filter (20 eV slit width). Recording was in super-resolution mode with a binned pixel size (equal to the physical pixel size) of 0.83 Å and dose-fractionated to 28 frames for a total exposure time of 1.4 s and a total dose of 45.3 e/Å$^2$. Raw cryo-EM movies were motion-corrected using UCSF MotionCor2 (*Zheng et al., 2017*) and CTF estimation was performed using Gctf (*Zhang, 2016*). Particle picking utilized a simple adversarial-template-based program DemoPicker written by FJS. Extracted particles were subjected to reference-

free 2D classification in RELION 3.1 and non-particles were removed. Rounds of 3D classification and refinement (C3 symmetry) were processed in RELION 3.1, using a cASIC1 cryo-EM map (emd_7009) as reference (*Yoder and Gouaux, 2020*). CTF refinement and Bayesian polishing were applied to further improve the resolution.

## Model building and refinement of hASIC1-Nb

Using a rigid-body fitting program, Molrep, the cryo-EM structure of the resting channel (PDB 6VTL) (*Yoder and Gouaux, 2020*) was docked into the cryo-EM density map. Docked models were used as templates for iterative rounds of manual model building in Coot (*Emsley et al., 2010*). For the Nb structure, a crystal structure of Nb (PDB 5IVO) was docked into the cryo-EM map by UCSF Chimera (*Goddard et al., 2007*) and was built in Coot. Additional real space refinement was performed using Refmac 5 (*Murshudov et al., 2011*) and Phenix (*Liebschner et al., 2019*).

## Immunofluorescence microscopy

HEK293T (ATCC CRL-11268) or SH-SY5Y (ATTC CRL-2266) were authenticated by the presence of low-pH induced currents sensitive to 50µM amiloride using patch clamp. Mycoplasma infection was ruled out by a commercial PCR assay (EZ-PCR Mycoplasma detection kit from Biological Industries, #20-700-20). Cells seeded onto glass coverslips treated with poly-L-lysine were either non-transfected or transfected with hASIC1a in pCDNA3.1 vector with Lipofectamine. For surface labeling of hASIC1a, 18–20 hr post-transfection, anti-hASIC1a Nbs were added to the culture medium in a concentration of 1 nM and cells were placed on ice to inhibit endocytosis. After 1 hr of incubation, cells were rinsed three times with PBS and fixed with 4% paraformaldehyde prepared in PBS for 30 min at 37°C. Cells were further permeabilized with buffer containing 1%Triton-X100 for 30 min. Mouse anti-Flag (Sigma) primary antibodies were added for 1 hr at RT. After three washes, Alexa Flour goat 594 and/or Alexa Flour 488 (Invitrogen) were added for 1 hr. Both primary and secondary antibody incubations were conducted in the presence of 5% normal goat serum to decrease background. DAPI was added to visualize nuclei. In chase experiments, Nb.C1 (1 nM) was incubated for 30 min, washed three times with PBS followed by a chase of 0, 1, 2, 3, and 4 hr at RT. At each time point, cells were fixed and processed for immunofluorescence. Coverslips were laid on glass slides with mounting solution (VECTASHIELD, Vector Laboratories H-1000). Images were captured with a Nikon confocal fluorescence microscope A1RMP LSM and analyzed using NIS Viewer 3.2. For visualizing intracellular hASIC1, cells were first fixed and permeabilized with 1% Triton-X100 before adding Nbs. The rest of the protocol was the same as described above. The human neuronal cell line SH-SY5Y (ATCC CRL-2266) was seeded on coverslips and treated with 10 nM staurosporin for 48 hr to induce cell differentiation. Cells were fixed, permeabilized prior to conducting immunofluorescence with Nb.C1-PcTx1 fusion as the primary antibody; the rest of the procedure was the same as above.

## Production of recombinant Nb.C1-PcTx1 polypeptides

The cDNA of Nb.C1 was modified first by introducing the DNA sequence of PcTx1 in the carboxyl-terminus of the Nb prior to the 6 His and HA epitopes using the unique restriction sites XhoI and XbaI. In a second step, a predicted flexible (KLGGGSGGGSAGSAAGGSGSGGEFGGGGSLE) or more rigid (GGGSGAEAAAKAEAEAKAEAAAKGGGGSG) linker was introduced using another pair of unique restriction sites: HindIII-XhoI, located upstream the PcTx1 cloning site. Plasmid was transformed into BL21(DE3) competent *E. coli* by heat shock. The previous protocol for purification of Nbs was followed with the following modifications. Cells were kept at 18°C during induction. Protease inhibitors and 1 mM β-mercaptoethanol were added to solutions of periplasmic protein purification. After affinity purification with Ni-NTA resin, proteins were concentrated using a Millipore concentrator and then added x2 Redox buffer (5 mL: 1.8 mg reduced glutathione added to 2.4 mg of oxidized glutathione resuspended in TN buffer without imidazole). His and HA tags were not removed. Peptides were further purified according to the protocol by *Saez et al., 2017*. Protein quantification was by absorbance at 280 nm using NanoDrop.

## Two-electrode voltage clamp of *Xenopus laevis* oocytes

Oocytes were injected with 5 ng of in vitro synthesized hASIC1a cRNA using the kit (mMESSAGEm-MACHINE T7, Thermo Fisher Scientific). Whole-cell currents were measured using a two-electrode

voltage clamp (Oocyte-Clamp OC-725C, Warner Instrument Corp.) with PowerLab 8/35 (ADInstruments) running LabChart Prosoftware. Electrode resistance was 0.5–1 MΩ when filled with 3 M KCl. Cells were placed in a fast-exchange perfusion chamber, <1 s, with high flow delivered by gravity. Perfusion solutions had the following composition (in mM): 100 NaCl, 4 KCl, 2 CaCl$_2$, 5 HEPES, 5 MES, pH was adjusted to desired values with N-methyl-D-glucamine. PcTx1 was used at 10 nM and MitTx at 50 nM, both were purchased from Alomone Labs. Oocytes were pre-incubated with purified Nb or Nb-linker-PcTx1 (10 nM) for 20 min at RT before recording. Oocytes were voltage-clamped at −60 mV and ASIC currents were elicited by fast change of the perfusion solution from pH 7.4 to 6.0 or as indicated in the experiments.

## Patch clamp of SH-SY5Y cells

Endogenous hASIC1 currents from human neuroblastoma cell line SH-SY5Y (ATCC CRL2266) were recorded in the whole-cell configuration using a HEKA patch clamp EPC10 amplifier and PATCH-MASTER acquisition software v2x90.2 (HEKA Electronic). Pipette solution contained in mM: 120 KCl, 20 HEPES pH 7.4. Bath solution contained 140 NaCl, 4 KCl, 2 CaCl$_2$, 20 HEPES adjusted to pH 7.4 or 6.0. Membrane potential was held at −60 mV. Cells were perfused with a solution of pH 7.4 to establish the baseline current, followed by activation with a solution of pH 6.0 using a fast-exchange perfusion system (SF-77B perfusion-step, Warner Instruments). Experiments were conducted at RT.

## Measurement of Nb.C1 affinity using a Biacore-S200 instrument

Purified hASIC1a was diluted in sodium acetate pH 5.0, 0.05% DDM to a concentration of 10 µg/mL. A chip (Series sensor Cytiva CM5) was activated by EDH/NHS (1-ethyl-3-[3-dimethylaminopropyl]carbodiimide)/(N-hydroxysuccinimide). Purified protein was injected, immobilized, and blocked by ethanolamine. In two experiments using different protein preps, the association time courses were recorded for 100 or 300 s, and dissociation for 300 or 3600 s, respectively. Flow rate=30µL/min. Nb.C1 was removed from the chip after testing each concentration using glycine pH 2.0 for 30 s, flow rate 30 µL/min. Binding data were generated by injecting the Nb.C1 at six concentrations (in nM: 0, 0.41, 1.23, 3.7, 11.1, 33.3, 100).

## Acknowledgements

We thank Chenghui Li for assistance with illustrations; Kaiyuan Ji for helping in the immunization and bleeding of the alpaca; Marc Llaguno for cryo-EM grid screening on the Glacios in the Yale Cryo-EM Core Facility; Shenping Wu and Kangkang Song, of the Yale Cryo-EM Core Facility and the University of Massachusetts Cryo-EM Core Facility, for Titan Krios data collection; and Prof. Wei Mi from the Department of Pharmacology at Yale University School of Medicine for advice on building the structural model.

## Additional information

### Funding

| Funder | Grant reference number | Author |
| --- | --- | --- |
| Beijing Advanced Innovation Center for Structural Biology, Tsinghua University | | Cecilia M Canessa |
| NIH | R21 MH10107466402 | Cecilia M Canessa |
| NIH | RO1 NS021501 | Fred Sigworth |

The funders had no role in study design, data collection and interpretation, or the decision to submit the work for publication.

### Author contributions

Yangyu Wu, Data curation, Formal analysis, Investigation, Methodology, Writing - original draft; Zhuyuan Chen, Formal analysis, Investigation, Methodology, Writing - original draft; Fred J Sigworth,

Software, Formal analysis, Supervision, Validation, Writing - review and editing; Cecilia M Canessa, Conceptualization, Supervision, Funding acquisition, Project administration, Writing - review and editing

### Author ORCIDs
Yangyu Wu (iD) https://orcid.org/0000-0001-8064-6132
Fred J Sigworth (iD) https://orcid.org/0000-0002-7178-8494
Cecilia M Canessa (iD) https://orcid.org/0000-0001-7316-5082

### Ethics
Animal experimentation: The study was carried out in accordance with the recommendations of the Institutional Animal Care Committee of Shanxi Agricultural University, China, where the alpaca was kept through the immunization protocol (protocol # 07990) The use of *Xenopus laevis* oocytes was approved by the Institutional Animal Care Committee of Tsinghua University, China (protocol #125154). Our group did not conduct frog surgeries, we received fragments of ovaries from the amphibian facility. The Association for Assessment and Accreditation of Laboratory Animal Care International (AALAC) has accredited Tsinghua amphibian animal facility.

### Decision letter and Author response
Decision letter https://doi.org/10.7554/eLife.67115.sa1
Author response https://doi.org/10.7554/eLife.67115.sa2

## Additional files
### Supplementary files
• Source data 1. ELISA screening 96-well plates of ELISA signal from screening of nanobodies recovered from third panning of library. In blue are indicated the clones with highest values.

• Source data 2. Values of oocyte currents elicited by changes in pH for activation and steady-state desensitization (SSD). Peak currents (μA/oocyte) elicited with solutions with the indicated pH measured with two-electrode voltage clamp. $pH_a$ = pH of activation, $pH_{ssd}$=pH of steady-state desensitization.

• Source data 3. Acid-induced currents (pH 6.0) in oocytes pretreated with nanobody (Nb). C1-Flexible linker-PcTx or Nb.C1-Rigid linker-PcTx shown if *Figure 5L*.

• Transparent reporting form

### Data availability
All data generated in this study are included in the MN and supporting files.

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
