## [Decision Letter]

**Acceptance summary:**

The work tackles a critical technical gap in the field of Acid-sensing ion channels: a specific antibody that marks human ASIC1a. The authors generate a nanobody specific to the extracellular part of human ASIC1a, which they use to obtain a high-resolution structure of the channel and modulate the binding of toxins to nearby sites on the channel. The nanobody will be a valuable tool for the structural, functional, and physiological characterization of ASIC1a, and is a launch point for the development of therapeutics that target pathologies of ASIC1a, including pain.

**Decision letter after peer review:**

Thank you for submitting your article "Structure and analysis of nanobody binding to the human ASIC1a ion channel" for consideration by *eLife*. Your article has been reviewed by 3 peer reviewers, including Randy B Stockbridge as the Reviewing Editor and Reviewer #1, and the evaluation has been overseen by Richard Aldrich as the Senior Editor. The following individual involved in review of your submission has agreed to reveal their identity: David M Maclean (Reviewer #2).

Essential revisions:

1) Authors should evaluate the current quality of the model (bonds, angles, C-β deviations, etc), perhaps use web servers like MolProbity for inspection. It is important to carry out these analyses until satisfactory model statistics are obtained. In addition, the authors should provide a detailed table of statistics of data collection and model refinement, as it provides one simple reference that readers can easily access. The table also serves as a guide for readers to assess the limitations of the presented data.

2) The current coordinate file is of one protomer with no symmetry operators to generate the other two protomers around the three-fold axis of symmetry. The authors described that RELION was used for 3D classification and refinement (C3 symmetry). Please provide symmetry operators.

3) For the parts of the manuscript where molecular interactions at the nanobody/channel interface are described, the authors should show maps, at least as a figure supplement.

4) For the data described in Table 1: some example raw traces and pH-response curves should be shown. In addition, some description should be provided of for variance in replicate measurements of pH dependence or the Hill coefficient.

5) All reviewers agreed that it would be helpful to the narrative of the manuscript to compare and contrast the binding sites for Nb.C1, MitTX, and PcTx1, perhaps by adding an additional figure. It would be helpful to add some discussion providing a structural rationalization of why Nb.C1 can bind hASIC1a in an overlapping binding site, but not alter function.

6) Data is not shown for certain assertions, such as that the signal for the hASIC1 with a deletion is decreased in ELISA assays (this was shown in immunofluorescence microscopy), or that the mouse ASIC1a shows weak interactions with the nanobody. Lines 139 and 189 refers to rat and mouse ASIC1a but that data is not shown.

7) Although recovery from the PcTx1 toxin is described as being more rapid than recovery from the fusion protein, the time constants of these processes are within 2-fold of each other (Figure 4K). The main difference between the recoveries is the plateau. One possible interpretation of this is that some of the fusion protein has been proteolyzed so that the recovery represents the behavior of two sub-populations: one of toxin without fused nanobody, and one with the fusion protein. Perhaps in agreement with this, the gel from the purification has lower molecular weight bands that may correspond to nanobody or toxin alone. The authors should discuss this possibility, and comment on whether any experiments were done to control for this.

8) The authors do not indicate whether the PcTx1-nanobody fusion is from multiple biochemical preparations or just a single protein prep. It is important to reproduce these results for toxin-fusions from multiple biochemical preps in order to understand the prep-to-prep reproducibility for measurements like the recovery plateau.

9) The authors focus on the C1 nanobody, but also mention results obtained with other nanobodies (the ones in the tree in Figure 1E). The authors state "Nb.C1 was selected on the basis of high-affinity, absence of modification of channel function and a profile of monodisperse hASIC1a protein in size exclusion…. (lines 106-108)". This suggests other nanobodies were screened for affinity, channel modification and peak dispersion. Also, lines 91-93, the authors mention other nanobodies which required permeabilization to bind. The data relating to affinity, channel modulation, SEC and immuno from other nanobodies should be included, not simply mentioned but never shown.

10) The authors should describe what criteria they use to identify nanobody Nb.C1 as "high affinity." Reporting a Kd value would be useful.

11) Lines 102-104, the authors mention hASIC1a tends to give low yields and aggregates, thus making cryo em sample prep difficult. The C1 nanobody is presented as a solution to this. The authors should include some data of hASIC1a alone (gels, chromatographs, etc.) to illustrate that this was a problem. As it stands, the nanobody is a solution to a problem we never actually see. Also, it is unclear how the Sun 2020 hASIC1a structure paper was able to overcome this without the benefit of nanobodies. Do the authors have thoughts on this?

12) The authors noted that SH-SY5Y was used for imaging as it contains endogenous hASIC1a. A figure is shown (Figure 4G) to demonstrate an inward current is elicited upon application of pH 6.0. Is this current amiloride-sensitive?

*Reviewer #1:*

The strengths of the manuscript are: This is the highest-resolution structure of human ASIC1 described to date. Often, nanobodies are selected and used for structural characterization without additional functional analysis; the focus on the functional consequences of this nanobody is refreshing, and the proposed applications of this nanobody are intriguing.

The weaknesses of the manuscript are:

Some data might require additional Table 1 does not indicate the variance in replicate measurements of pH dependence or the Hill coefficient for the channel. Data is not shown for certain assertions, such as that the signal for the hASIC1 with a deletion is decreased in ELISA assays (although this was shown in immunofluorescence miscroscopy), or that the mouse ASIC1a shows weak interactions with the nanobody. Although the nanobody is described as high-affinity, the authors do not report a Kd value. Given that the foundation of this manuscript is characterization of nanobody-channel interactions, this seems like an important piece of information.

Although recovery from the PcTx1 toxin is describes as being more rapid than recovery from the fusion protein, the time constants of these processes are within 2-fold of each other (Figure 4K). The main difference between the recoveries is the plateau. The authors do not speculate about what this might mean, mechanistically. One possibility is that some of the fusion protein has been proteolyzed so that the recovery represents the behavior of two sub-populations: one of toxin without fused nanobody, and one with the fusion protein. Perhaps in agreement with this, the gel from the purification has lower molecular weight bands that may correspond to nanobody or toxin alone. Related to this, the authors do not indicate whether the PcTx1-nanobody fusion is from multiple biochemical preparations or just a single protein prep. It is important to reproduce these results for toxin-fusions from multiple biochemical preps in order to understand the prep-to-prep reproducibility for measurements like the recovery plateau.

*Reviewer #2:*

This manuscript describes the creation and characterization of a nanobody directed at the human ASIC1a extracellular domain. A nanobody library was generated by injecting an alpaca with cells expressing human ASIC1a (hASIC1a) then extracting mRNA from peripheral lymphocytes and using this to create a phage display library. The phage display library went through three different rounds of screening to arrive at a pool of about 20 promising clones. One of these was selected for further detailed characterization. This particular nanobody (C1) was able to stabilize trimeric hASIC1a in size exclusion, enabling the authors to obtain high resolution cryo-EM structures of nanobody-bound hASIC1a. This structure indicates the nanobody binds to an extracellular loop which in hASIC1a contains an added two amino acids compared to other ASIC1 variants, providing a mechanism for specificity. The evidence for specificity is largely provided by immunofluorescence data with either a control nanobody or modified ASIC subunits as controls. Interestingly, the C1 nanobody does not appear to alter hASIC1a function but does compete with the peptide MitTx, which binds nearby. Conjugating the nanobody to an ASIC-selective toxin can considerably prolong the toxins dissociation. The properties of this specific nanobody hold promise as a useful tool in the field.

The major advance is isolating this hASIC1a specific nanobody. Based on the authors data, this nanobody is specific due to two amino acids added to an extracellular loop in hASIC1a which are not present in chicken ASIC1 or other ASICs. As stated by the authors, this nanobody could then be used to help structural and biochemical investigation of hASIC1a, or potentially for therapeutic purposes in the future. A limitation or weakness of this nanobody is it only seems to work for hASIC1a, not mouse or rat ASIC1a. Nonetheless, the approach is strong, the data appear clear and support the major claims of the paper. There are a number of occasions where more detail is warranted, or where additional controls would increase overall quality but nothing that threatens the core findings of the manuscript.

*Reviewer #3:*

Wu et al. identify a novel alpaca-derived nanobody (Nb.C1) that binds specifically and with high affinity for human acid-sensing ion channel 1a (hASIC1a) and does not alter the functional properties of the channel. The authors use a combination of cryo-EM, electrophysiology, and fluorescence-based imaging to probe the hASIC1a-Nb.C1 complex. The authors demonstrate that by using Nb.C1, they were able to overcome biochemical challenges related to isolation of hASIC1a and were able to obtain a 3D map at 3.15 Å, a marked improvement from a previous cryo-EM structure of hASIC1a resolved at 3.9 Å by Sun et al. (2020). The resulting map shows hASIC1a in complex with three Nb.C1 in the closed, resting state. Interestingly, Nb.C1 binding site overlaps with that of MitTx and Mambalgin-1, toxins that stabilized the open and closed states of ASIC1a, respectively, but not that of PcTx1. The authors probe the interactions of Nb.C1 with hASIC1a using MitTx as a functional marker as it activates ASIC1a at physiological pH by electrophysiology, and the functional results support the structure. Application of both Nb.C1 and MitTx reduces the effects of the toxin. Wu et al. also investigated the effects of the Nb.C1 fused with PcTx1 and found that the fusion resulted prolonged effects of PcTx1-mediated inhibition. Altogether, the authors conclude that Nb.C1 is a powerful tool with multiple applications serving both structural-based and therapeutic-based studies. Indeed, the conclusion of the work described are largely supported by the data presented, but there are multiple sections in the manuscript that authors can expand on, clarify, and thoroughly inspected for consistency to strengthen the work.

1) A table of statistics relating to data collection and model refinement would be greatly appreciated as it provides one simple reference that readers can easily access. The table also serves as guidance for readers in order to assess the limitations of the presented data.

2) It is striking that the Nb.C1 binding site overlaps those of MitTx and Mambalgin-1 but the nanobody does not demonstrate either activating or inhibiting effects on the functional properties of hASIC1a. Authors should expand on the similarities and differences between the binding sites of Nb.C1 and toxins as this highlights the utility of Nb.C1 for different applications, as the authors suggest. Furthermore, the binding site differences might shed light into how Nb.C1 can specifically bind to hASIC1a without altering its function, unlike the binding of toxins.

[Editors' note: further revisions were suggested prior to acceptance, as described below.]

Thank you for resubmitting your work entitled "Structure and analysis of nanobody binding to the human ASIC1a ion channel" for further consideration by *eLife*. Your revised article has been evaluated by Richard Aldrich as the Senior Editor, and a Reviewing Editor.

The manuscript has been improved but there are some remaining issues that need to be addressed, as outlined below:

1. In Figure 5K, the datapoints obtained from the two constructs (rigid and flexible linker) should not be combined, since the reader cannot evaluate whether the constructs behave similarly. These should be shown as separate datasets in Figure 5K.

2. Performing a second biochemical purification of the nanobody/toxin fusion is important to evaluate the prep-to-prep reproducibility of the quantitative values reported for Figure 5J and 5K (point #8 in the original review). It needs to be established whether the fraction of current that is recovered is something that is consistent (and may be a property of the chimeric molecule being tested), or variable from prep-to-prep (and may be due to contaminating toxin not covalently linked to monobody). Related to point #1, separating the data obtained from the rigid and flexible linker could help address whether similar time constants and plateaus are observed for toxin-nanobody chimeras that have been prepared in independent batches (with the caveat that the constructs are not exactly the same). As an alternative, these data could be removed from the manuscript.

3. The newly presented Biacore data for the binding of the nanobody to the channels (shown in Figure 5-S1) is not well fit by the two-site binding model that is used. This fit leads to an exceptionally low 1 pM binding affinity, which is likely an overestimate of the binding affinity. (As a frame of reference, "tight" antibody-antigen complexes are usually closer to 1 nM binding affinity, with some as low as 100 pM). These data should not be used to estimate a Kd value if issues with protein quality make these curves unfittable by plausible binding models.

---

## [Author Response]

Essential revisions:1) Authors should evaluate the current quality of the model (bonds, angles, C-β deviations, etc), perhaps use web servers like MolProbity for inspection. It is important to carry out these analyses until satisfactory model statistics are obtained. In addition, the authors should provide a detailed table of statistics of data collection and model refinement, as it provides one simple reference that readers can easily access. The table also serves as a guide for readers to assess the limitations of the presented data.

Thank you for the suggestions, we have carried out the improved refinement and evaluation. We have added the table as a Figure 2-Table supplement 2.

2) The current coordinate file is of one protomer with no symmetry operators to generate the other two protomers around the three-fold axis of symmetry. The authors described that RELION was used for 3D classification and refinement (C3 symmetry). Please provide symmetry operators.

The symmetry operators are listed below.

**Author response table 1. resptable1:** 

	Symmetry type	Symmetry fold	X-axis element	Y-axis element	Z-axis element	Angle (degrees)	Peak height
Symmetry axis #1	C	3	-0.01424	-0.02216	0.9997	120	0.7364

3) For the parts of the manuscript where molecular interactions at the nanobody/channel interface are described, the authors should show maps, at least as a figure supplement.

We have added to Figure 3C the map density of the interface between Nb.C1 and hASIC1a. See Figure 3-C.

4) For the data described in Table 1: some example raw traces and pH-response curves should be shown. In addition, some description should be provided of for variance in replicate measurements of pH dependence or the Hill coefficient.

i) We have added representative examples of hASIC1a with Nb-GFP (control Nb directed against GFP) and hASIC1a with 10 nM Nb-C1 showing response to proton activation and steady state desensitization: Figure 3—figure supplement 5.

ii) Table I shows mean± SD values for all measurements. The original data is included in an Excel file as Source Data 2.

iii) Also added are representative traces and results of pH_50a_ and pH_50ssd_ of hASIC1a wildtype and hASIC1a-DDL and cASIC wildtype and cASIC+DL. The purpose of these experiments was to test whether residues DL have any functional effect on ASICs. The results show that deletion of DL in hASIC1a produces a shift of 0.08 pH units of the pH_50ssd_ to more alkaline value (t-test p=0.008) whereas adding DL to cASIC shifts the pH_50ssd_ by 0.07 pH units (p=0.005) to more acid value (Sherwood and Askwith, 2008). The raw data has been included in Source Data 2.

5) All reviewers agreed that it would be helpful to the narrative of the manuscript to compare and contrast the binding sites for Nb.C1, MitTX, and PcTx1, perhaps by adding an additional figure. It would be helpful to add some discussion providing a structural rationalization of why Nb.C1 can bind hASIC1a in an overlapping binding site, but not alter function.

i) We have added new paragraphs to the text (Lines 162-178) comparing the binding sites.

ii) We have added a new paragraph discussing the lack of functional changes (lines 293-296).

6) Data is not shown for certain assertions, such as that the signal for the hASIC1 with a deletion is decreased in ELISA assays (this was shown in immunofluorescence microscopy), or that the mouse ASIC1a shows weak interactions with the nanobody. Lines 139 and 189 refers to rat and mouse ASIC1a but that data is not shown.

In the first submission we showed IF with strong signal for hASIC1a wildtype but no signal when residues DL are deleted, and no signal with cASIC1. We have added to Figure 3—figure supplement 3 new IF confocal images of mASIC1a, mASIC2a and mASIC3 with Nb.C1. The new images show no reactivity with any of those channels, indicating high specificity of Nb.C1 to hASIC1a.

The sentence mentioning ELISA has been removed because we didn’t conduct purification of cASIC, mASIC1a, ASIC2a or mASIC3 proteins that are necessary for the respective ELISA assays. Instead, we used IF to demonstrate specificity of the Nbs.

We include results of ELISA in Source Data 1: raw data from 96-well plate readings that include Nbs C1 to H10.

7) Although recovery from the PcTx1 toxin is described as being more rapid than recovery from the fusion protein, the time constants of these processes are within 2-fold of each other (Figure 4K). The main difference between the recoveries is the plateau. One possible interpretation of this is that some of the fusion protein has been proteolyzed so that the recovery represents the behavior of two sub-populations: one of toxin without fused nanobody, and one with the fusion protein. Perhaps in agreement with this, the gel from the purification has lower molecular weight bands that may correspond to nanobody or toxin alone. The authors should discuss this possibility, and comment on whether any experiments were done to control for this.

The interpretation provided by the reviewers is very likely. In addition, it is possible that a fraction of PcTx is not well folded i.e., pairing of the six cysteines is incorrect diminishing toxin binding affinity of the fusion protein. We have added an explanation to the text starting at line 224.

8) The authors do not indicate whether the PcTx1-nanobody fusion is from multiple biochemical preparations or just a single protein prep. It is important to reproduce these results for toxin-fusions from multiple biochemical preps in order to understand the prep-to-prep reproducibility for measurements like the recovery plateau.

We were aware of this problem. Because our limited resources, the purification of each fusion protein was conducted only once. The amount of purified fusion proteins was sufficient for the functional experiments. This caveat is mentioned in the legend of Figure 5.

9) The authors focus on the C1 nanobody, but also mention results obtained with other nanobodies (the ones in the tree in Figure 1E). The authors state "Nb.C1 was selected on the basis of high-affinity, absence of modification of channel function and a profile of monodisperse hASIC1a protein in size exclusion…. (lines 106-108)". This suggests other nanobodies were screened for affinity, channel modification and peak dispersion. Also, lines 91-93, the authors mention other nanobodies which required permeabilization to bind. The data relating to affinity, channel modulation, SEC and immuno from other nanobodies should be included, not simply mentioned but never shown.

Actually, we did not screen all isolated clones by all the criteria listed above, that approach would have taken a lot of time and resources.

After the third panning, we used ELISA to identify the strongest positive phages. As indicated in Figure 1D, an arbitrary threshold was set to examine only phages with high intensity signal, others were not considered for the next step. Several dozens of clones highly positive in ELISA were selected for DNA sequencing. Most of the sequenced clones were repeats or they had very similar sequences; i.e., there were not dozens of different clones rather a few clones repeated many times. This result is expected and represents a successful selection by our panning strategy of high affinity clones versus weak binders. Some of the sequenced clones were eliminated because they did not produce high amount of protein. The finalist clones are shown in Figure 1E. We next used IF to select clones with reactivity to hASIC1a and low background. The group in blue gave the cleanest signal. The group in green was also very clean but because the epitope(s) is intracellular we did not pursue further characterization in this study. If one looks at the sequences of the blue group (Figure 1 supp 2), they are almost identical, they differ by only one or two residues. Members of this group were tested on channels expressed in cells by TEVC. We used first C1 and later D10 in hASIC1a purification. Both Nbs were very good for this task, so we settled for C1 for the rest of the study.

The original sentence has been modified to convey a correct summary of the criteria that finally selected Nb.C1 though is not an exact sequence of events in the selection process:

“Among the best binders initially screened, Nb.C1 was selected on the basis of high-affinity, low background, and absence of modification of channel function. Subsequently, Nb.C1 was added to large scale preparations of crude membranes…”

It is entirely possible, indeed very likely, that the library contains more ‘good nanobodies’ for other purposes; they could be isolated by modifying the panning strategy.

Regarding Nbs in the green group, we are pursuing other applications, but for this publication is irrelevant. For the moment the library remains in the freezer until we obtain funds to continue the work.

10) The authors should describe what criteria they use to identify nanobody Nb.C1 as "high affinity." Reporting a Kd value would be useful.

As explained in the answer to the previous question, a good strategy for screening the phage library is the best way to isolate high affinity and specific binders. We have conducted measurements of binding kinetics of Nb.C1 using SPR with a Biacore instrument (Figure 5-supplement 1)

11) Lines 102-104, the authors mention hASIC1a tends to give low yields and aggregates, thus making cryo em sample prep difficult. The C1 nanobody is presented as a solution to this. The authors should include some data of hASIC1a alone (gels, chromatographs, etc.) to illustrate that this was a problem. As it stands, the nanobody is a solution to a problem we never actually see. Also, it is unclear how the Sun 2020 hASIC1a structure paper was able to overcome this without the benefit of nanobodies. Do the authors have thoughts on this?

In our hands, purification of high quality hASIC1a expressed in HEK293F cells was challenging owing to aggregation, and when placed on grids, displayed strong orientation bias. Nb.C1 solve both problems. Representative examples of SEC profiles using 1% DDM or 1% Fos-choline14 are shown, along with a Western blot of hASIC1a protein from the three peaks as Figure 2—figure supplement 1. In the figure legend we note the differences between a well-behaved protein of Sun et al. and our protein.

12) The authors noted that SH-SY5Y was used for imaging as it contains endogenous hASIC1a. A figure is shown (Figure 4G) to demonstrate an inward current is elicited upon application of pH 6.0. Is this current amiloride-sensitive?

Yes. Noted in the figure legend.

[Editors' note: further revisions were suggested prior to acceptance, as described below.]

The manuscript has been improved but there are some remaining issues that need to be addressed, as outlined below:1. In Figure 5K, the datapoints obtained from the two constructs (rigid and flexible linker) should not be combined, since the reader cannot evaluate whether the constructs behave similarly. These should be shown as separate datasets in Figure 5K.2. Performing a second biochemical purification of the nanobody/toxin fusion is important to evaluate the prep-to-prep reproducibility of the quantitative values reported for Figure 5J and 5K (point #8 in the original review). It needs to be established whether the fraction of current that is recovered is something that is consistent (and may be a property of the chimeric molecule being tested), or variable from prep-to-prep (and may be due to contaminating toxin not covalently linked to monobody). Related to point #1, separating the data obtained from the rigid and flexible linker could help address whether similar time constants and plateaus are observed for toxin-nanobody chimeras that have been prepared in independent batches (with the caveat that the constructs are not exactly the same). As an alternative, these data could be removed from the manuscript.

Answer to the above two comments. We made new preparations of fusion proteins (Nb.C1-FlexLinker-PcTx and Nb.C1-RigidLinker-PcTx) and repeated the k_off_ measurements in oocytes with TEVC. The new data are presented in Figure 5K-L. There is a difference in the maximal inhibition (16% and 14%) between the two constructs. The Coomassie blue SDS-PAGE shows (Figure 5J) a small fraction of cleaved protein -more prominent in the flexible linker fusion- that we believe may account for the more rapid than expected koff, as it generates free toxin. Though we optimized the purification protocol for these preps (lower temperature during induction, shorter incubation, addition of protease inhibitors and small concentration of reducing agents, indicated in methods), the cleavage was not completely eliminated. This issue could be solved by trying different sequences of linkers resistant to proteolysis, though at the expense of significant additional time. We provide here stronger evidence that the fusions potentiate the effect of PcTx1 by producing a large non-recovering fraction.

3. The newly presented Biacore data for the binding of the nanobody to the channels (shown in Figure 5-S1) is not well fit by the two-site binding model that is used. This fit leads to an exceptionally low 1 pM binding affinity, which is likely an overestimate of the binding affinity. (As a frame of reference, "tight" antibody-antigen complexes are usually closer to 1 nM binding affinity, with some as low as 100 pM). These data should not be used to estimate a Kd value if issues with protein quality make these curves unfittable by plausible binding models.

We have removed the figure of the Biacore binding kinetics of Nb.C1, as we agree that the fit to the unbinding time-course is very unreliable. The following paragraph has been added describing how we derived a rough estimate of the K_D_ value of Nb.C1 (lines 214-222):

“To estimate the nanobody binding affinity, we first measured binding of Nb.1C to immobilized, detergent-solubilized hASIC1a protein using surface plasmon resonance. From experiments with two different Nb and hASIC1a protein preparations, global fits to association time courses with nanobody concentrations from 0.4 to 100nM yielded association rate constants k_on_ ranging from 6.8x10^4^ to 2.2x10^5^ M^-1^ s^-1^. […] Taking the smallest k_on_ value as a lower bound on association rate yields the equilibrium constant K_d_ ≲ 0.2 nM.”

This value is in the range of llama monomeric nanobodies (doi.org/10.1038/s42586-021-03676-z).